# M²-VLP: Enhancing Multilingual Vision-Language Pre-Training via Multi-Grained Alignment

## Abstract

Recently, multilingual Vision-Language Pre-training (mVLP) has shown remarkable progress in learning joint representations across different modalities and languages. However, most existing methods learn semantic alignment at a coarse-grained level and fail to capture fine-grained correlations between different languages and modalities. To address this, we propose a **M**ulti-grained **M**ultilingual **V**ision-**L**anguage **P**re-training (M²-VLP) model, which aims to learn cross-lingual cross-modal alignment at different semantic granular levels. In cross-lingual interaction, the model learns the global alignment of parallel sentence pairs and the word-level correlations. In cross-modal interaction, the model aligns images with captions and image regions with corresponding words. To integrate the cross-lingual and cross-modal alignment above, we propose a unified multi-grained contrastive learning paradigm. Under zero-shot cross-lingual and fine-tuned multilingual settings, extensive experiments on vision-language downstream tasks across twenty languages demonstrate the effectiveness of M²-VLP over competitive contrastive models. The anonymous code is available in https://anonymous.4open.science/r/M2-VLP-ANNO-27BA.

## CCS Concepts

• **Computing methodologies → Natural language processing**.

## Keywords

Multilingual vision-language pre-training, Multi-modal alignment, Cross-lingual Transfer

**ACM Reference Format:**

Anonymous Author(s). 2025. M²-VLP: Enhancing Multilingual Vision-Language Pre-Training via Multi-Grained Alignment. In *Proceedings of the ACM Web Conference (WWW '25)*. ACM, New York, NY, USA, 12 pages. https://doi.org/XXXXXXX.XXXXXXX

## 1 Introduction

Vision-Language Pre-training (VLP) research plays a pivotal role in shaping the future landscape of the internet, as it greatly enhances the intelligence of web applications (e.g., search engines [42] and recommendation [45]). Typically, VLP models learn cross-modal joint representations from large-scale image-text pairs during the pre-training stage, which are then fine-tuned on downstream vision-language tasks, such as visual question answering,

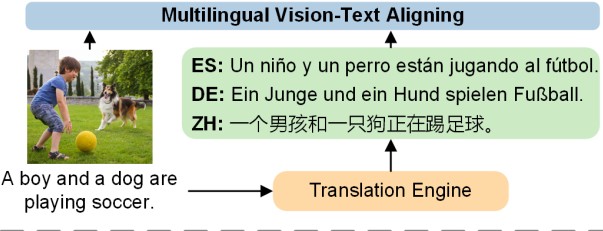

(a) Translation-based Methods

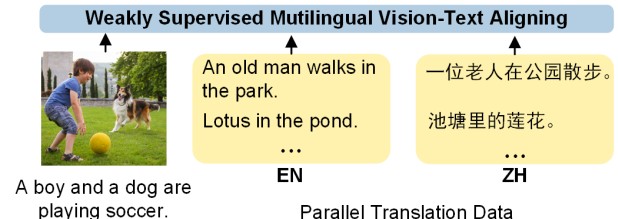

(b) Unified Modeling Methods

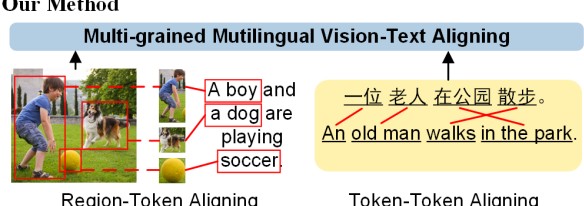

(c) Our Method

**Figure 1: An illustration of the differences between existing mVLP methods and M²-VLP. Most existing methods align cross-lingual cross-modal representations at a coarse-grained level, while M²-VLP performs multi-grained aligning.**

image captioning, and image-text retrieval [7]. While most works are based on English, recent studies on multilingual VLP (mVLP) have attempted to overcome the language barrier by extending VLP models to multilingual scenarios. By aligning the representations from different modalities and languages, mVLP models achieve promising results on various cross-lingual cross-modal downstream tasks[6, 30, 34, 52].

However, most existing mVLP methods only learn cross-lingual and cross-modal alignment at a coarse-grained level, ignoring the critical role of explicit fine-grained semantic alignment. As illustrated in Figure 1 (a, b), previous methods can be categorized mainly into two paradigms: (a) The translation-based methods [6, 34, 52] extend English image-text pairs to other languages using translation engines, subsequently performing VLP on the generated multilingual data. (b) The unified modeling methods [30] aim to

align cross-modal image-text pairs and cross-lingual translation pairs within a unified framework. Despite their advances, the pre-training paradigms of these models are limited to coarse-grained level, lacking explicit supervision to capture fine-grained correlations between different languages and modalities.

In this paper, we explore cross-lingual cross-modal alignment from the perspective of human learning. When we learn a new language, the initial step is to memorize the vocabulary and align the vocabulary with our native language. Similarly, when young children learn to recognize images, they are first taught to associate the objects depicted in the image with the corresponding concepts. These observations suggest that fine-grained token-token and region-token aligning signals may be effective in understanding detailed correlations in both cross-lingual and cross-modal interaction. Inspired by this, we introduce a Multi-grained Multilingual Vision-Language Pre-training model, namely $M^2$-VLP. As shown in Figure 1 (c), the model integrates four aligning tasks with different granular levels through a unified multi-grained contrastive learning strategy. In cross-modal contrastive, $M^2$-VLP aligns global images with English captions at a coarse-grained level, and image regions with corresponding English phrases at a fine-grained level. In cross-lingual contrastive, the model simultaneously learns sentence-level and token-level alignment using parallel translation pairs. Through a unified formulation for different data streams, the model achieves multi-grained cross-lingual cross-modal cascading alignment without any multilingual image-text pairs. To further improve cross-lingual and cross-modal interactions, $M^2$-VLP performs three Masked Language Modeling (MLM) tasks, including Vision MLM (VMLM), cross-lingual MLM (xMLM), and Translation Language Modeling (TLM).

Under zero-shot cross-lingual and fine-tuned multilingual settings, we demonstrate the effectiveness of $M^2$-VLP across a broad range of downstream vision-language tasks, including visually-grounded natural language inference, visual question answering, visual reasoning, and image-text retrieval. Our experiments cover a set of 20 target languages from diverse language families. Experimental results show that $M^2$-VLP significantly outperforms competitive mVLP models with an averaged improvement of 2.9% and 10.6% on cross-lingual vision-language understanding and retrieval tasks in the IGLUE benchmark. Visualization analysis of the proposed model further demonstrates its ability to perform fine-grained cross-modal and cross-lingual interactions.

Our contributions can be summarized as follows:

- We present $M^2$-VLP, the first known effort to explicitly learn multi-grained cross-lingual and cross-modal aligning in multilingual vision-language pre-training.
- A unified contrastive learning strategy is proposed to learn Vision-to-English and English-to-X languages alignment in a multi-grained manner. With English serving as the aligning bridge, it achieves cross-lingual cross-modal aligning without any multilingual image-text data.
- Extensive experiments on 7 vision-language tasks across 20 languages demonstrate the effectiveness of $M^2$-VLP. Further analysis is conducted to show the fine-grained aligning ability across different modals and languages.

## 2 Related Works

### 2.1 Multilingual Language Models

Recent studies [10, 11, 13] have demonstrated the effectiveness of multilingual pre-trained language models on various downstream tasks. Multilingual BERT [13] is the first work to extend the monolingual pre-training to the multilingual setting by performing masked language modeling (MLM) on large-scale multilingual corpora. XLM [11] introduced translation language modeling to achieve better cross-lingual alignment, while XLM-R [10] further enlarged the model size and training corpora. Based on these models, several methods have been proposed to enhance multilingual representation through different perspectives. InfoXLM [8] introduces sentence-level contrastive loss, aiming to maximize the mutual information between translation pairs. HICTL [46] bridges the semantic discrepancy across languages through hierarchical contrastive learning. UniPropmt [17] introduces a language-agnostic prompting model to alleviate the effort of designing multilingual prompt templates for different languages. MLM-GC [2] leverages the global co-occurrence information from multilingual corpora to enhance semantic alignment. EMMA-X [16] integrates cross-lingual representation learning with semantic relation prediction within an expectation-maximization framework. In addition, several studies focus on improving language-agnostic representations [9] and mitigating the influence of linguistic discrepancy [23] between the source and target languages.

### 2.2 Vision-Language Pre-training

Based on the way of integrating features from vision and language, VLP models generally fall into two categories: dual-encoder and fusion-encoder architecture.

The dual-encoder model [20, 36] consists of an image encoder and a text encoder to encode images and text separately. Then, it adopts straightforward methods such as shallow attention layer [24] or dot product [20, 36] to model the interaction between different modalities. However, the simple interaction is not enough to handle tasks that require complex reasoning, such as visual reasoning and visual question answering [19].

The fusion-encoder model takes text embeddings and image features as input and employs a deep fusion encoder to learn vision-language interaction. VisualBERT [27] and VL-BERT [41] implicitly align elements of an input text and associated image regions with self-attention. Instead of simply using image-text pair, OS-CAR [29] incorporates object detection tags within the image to enhance the fusion encoder to better align different modalities. ALBEF [26] utilizes a text and a vision encoder to independently learn intra-modal interaction, followed by a fusion-encoder with cross-attention for cross-modal interaction. BEIT-3 [44] proposes a multiway transformer by performing masked data modeling on images, texts, and image-text pairs. Besides, some approaches leverage the pre-trained Large Language Models (LLM) to enhance VLP. For example, Flamingo [3] aligns the vision encoder and LLM using a perceiver resampler, which shows remarkable few-shot performance. BLIP-2 [25] bridges the modality gap with a lightweight querying transformer. LLAVA [33] connects the visual encoder of CLIP [36] with the language decoder Vicuna [50], and conducts end-to-end fine-tuning on generated instructional vision-language

dataset. Due to the scarcity of large-scale multilingual image-text data, most VLP methods mainly focus on English.

## 2.3 Multilingual Vision-Language Pre-training

As multilingual language models advance rapidly, some research attempts to explore universal representations across multiple languages and modalities. MURAL [18] extends the ALIGN [20] model to multilingual settings by applying multi-task contrastive learning on image-text pairs and translation pairs. To construct multilingual image-text pairs, M-CLIP [6] and UC$^2$ [52] extend English-only datasets via machine translation, and leverage the generated datasets for vision-language alignment. As these methods highly depend on the quality of the translation engine, some research has attempted alternative paradigms. RC$^3$ [51] applies regularized contrastive learning that constrains the representation proximity of weakly-aligned multilingual vision-language inputs. M$^3$P [34] enforces explicit alignment through a code-switching strategy, replacing English words in image-text pairs with their synonyms in the target languages. Li et al. [30] proposes a weakly supervised framework to effectively unify cross-lingual and cross-modal pre-training, achieving remarkable results on various cross-lingual vision-language tasks.

## 3 Methodology

In this section, we present a multi-grained multilingual framework M$^2$-VLP. We first briefly introduce the three types of data streams used for pre-training in section 3.1. Then we describe the model architecture and pre-training objectives in section 3.2 and 3.3 respectively.

### 3.1 Data Stream

We use three data streams: multilingual text stream, parallel text stream, and monolingual image-text stream. Different from most previous methods, our approach does not rely on machine translation engines or multilingual image-text datasets. Details regarding the three data streams are as follows:

**Multilingual Text Stream**    To learn cross-lingual modeling, we use multilingual text stream as model input. Given $N$ languages $\{L_i\}_{i=1}^N$, we construct a multilingual text dataset $D_M = \cup_{i=1}^N \{x_j^{L_i}\}_{j=1}^{N_{L_i}}$, where $N_{L_i}$ denotes the number of training sentences in language $L_i$ and $x_j^{L_i}$ is the $j$-th sentence in language $L_i$. Due to significant differences in data size for different languages, sentences are sampled according to a multinomial distribution [10]. This sampling increases the proportion of low-resource languages and alleviates the training bias towards high-resource languages.

**Parallel Text Stream**    To learn cross-lingual semantic alignment, we apply parallel text stream. Given $N$ languages $\{L_i\}_{i=1}^N$, we construct an English-X parallel text dataset $D_T = \cup_{i=1}^N \{x_j^{En}, y_j^{L_i}\}_{j=1}^{N'_{L_i}}$, where $\{(x_j^{En}, y_j^{L_i})\}$ is the $j$-th English-$L_i$ translation pair and $N'_{L_i}$ denotes the number of translation pairs in $L_i$.

**Monolingual Image-Text Stream**    To learn cross-modal modeling, we use monolingual image-text stream. This data stream consists of two parts: coarse-grained image-caption data and fine-grained region-token data. We denote image-caption data as $D_I =$

$\{(I_j, x_j^{En})\}_{j=1}^{N_I}$, where $(I_j, x_j^{En})$ represents an image-caption pair and $N_I$ is the number of image-caption samples. For images in $D_I$, we also construct fine-grained region-token dataset $D_R = \{(R_j, x_j^{En})\}_{j=1}^{N_R}$, where $(R_j, x_j^{En})$ denotes an region-token pair and $N_R$ is the number of region-token samples. Note that the "token" here can not only be a word, but also textual phrases that describe a specific region of the image. Specifically, we utilize the existing image annotations of object detection and region description to form region-token pairs. Based on region and object coordinate labels, we cut the original images to obtain region $R_j$. $x_j^{En}$ for objects are original object labels. If an object annotation contains additional attributes (e.g., color or shape), we concatenate the attributes with the original labels as the region description. $x_j^{En}$ for regions are phrases that describe the specific regions in original images. Furthermore, we augment $D_I$ and $D_R$ with a code-switching strategy, in which the model randomly replaces tokens in image-text pairs with their target language synonyms.

### 3.2 Model Architecture

As illustrated in Figure 2, we construct a two-steam framework that contains a vision encoder to learn visual features, a text encoder to learn textual features, and a fusion encoder to learn cross-modal interactions. The text and vision encoders adopt standard Transformer [13] architecture with $N_V$ and $N_T$ layers respectively.

Specifically, texts and images are fed into the corresponding uni-modal encoders to perform intra-modal interaction. Since our method is based on multi-grained aligning, "image" can refer to a global image or a specific region, and "text" can refer to a complete caption or a text phrase describing the corresponding region. For text input, we feed it to the text encoder and get the text representation $T = \{t_{cls}, t_1, ..., t_n\}$. For image input, the vision encoder transforms images into fixed-size $m$ patches to get image representation $I = \{i_{cls}, i_1, ..., i_m\}$. Similar to language models [13], a special token [CLS] is prepended to the image patches, serving as the representation of the global image.

After that, image and text representations are fed into the fusion encoder. To learn the alignment between different modalities and languages, we follow [30] to adopt a pluggable cross-attention layer in the fusion module, which allows different routines for different data streams. For image-text stream $D_I$ and $D_R$, the cross-attention layers are activated to learn cross-modal interaction. Mathematically, the cross-attention between text feature $x_t$ and image feature $x_i$ can be denoted as

$$q = x_t \mathbf{W}_q, \quad k = x_i \mathbf{W}_k, \quad v = x_i \mathbf{W}_v,$$

$$\text{Attention}(q, k, v) = softmax(\frac{qk^T}{\sqrt{d_k}}) \cdot v, \quad (1)$$

where $\mathbf{W}_q \in \mathbb{R}^{d_m \times d_q}$, $\mathbf{W}_k \in \mathbb{R}^{d_m \times d_k}$, and $\mathbf{W}_v \in \mathbb{R}^{d_m \times d_v}$ are learnable weight matrices, and $d_m, d_q, d_k,$ and $d_v$ are dimensions of the input embedding, query, key, and value vectors. For parallel text stream $D_T$, the cross-attention layers are activated to learn cross-language interaction. The cross-attention query, key, and value vectors of parallel text input $(x_p, x_q)$ can be formulated as

$$q = x_p \mathbf{W}_q, \quad k = x_q \mathbf{W}_k, \quad v = x_q \mathbf{W}_v. \quad (2)$$

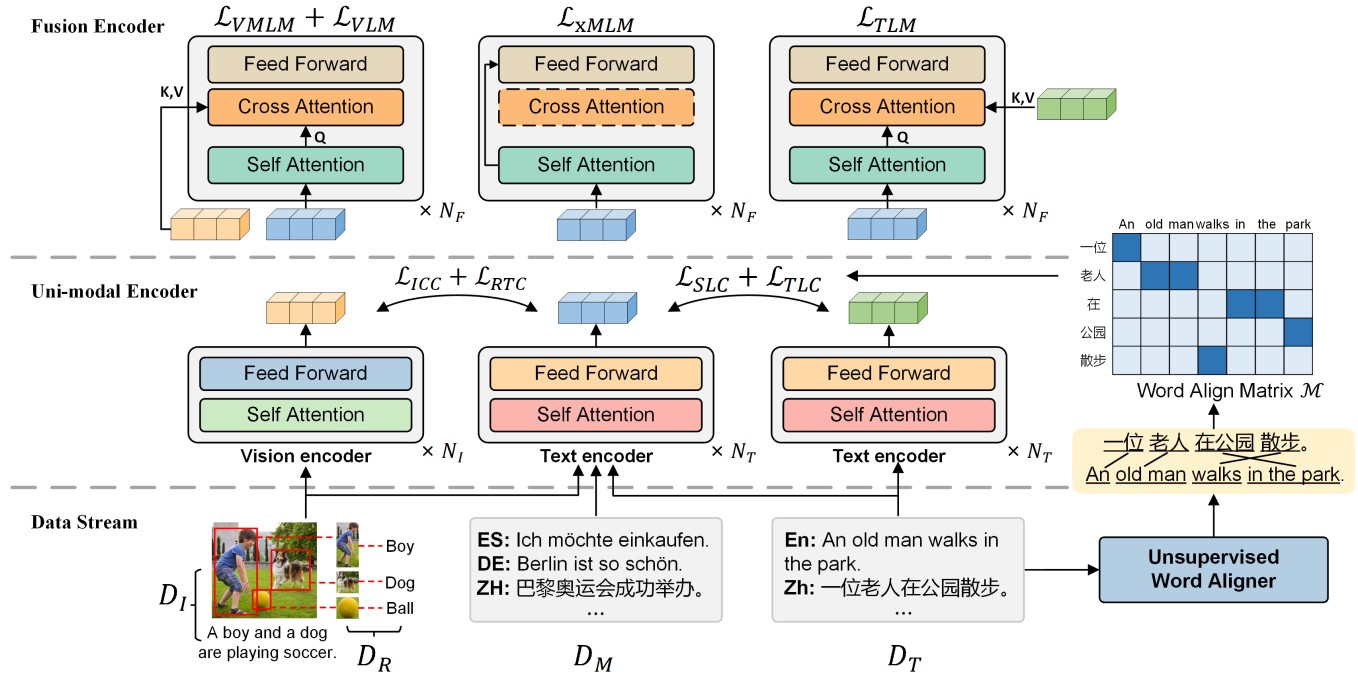

**Figure 2: Illustration of the proposed framework. M²-VLP consists of vision, text, and fusion encoders. Modules with the same color share the same parameters. We use cuboids with different colors to highlight the data flow within the model. $\mathcal{L}$ with different subscripts denotes different pre-training losses.**

For unpaired text stream $D_M$, the cross-attention layers are skipped. Different from previous methods that learn cross-modal and intra-modal modeling in a single self-attention layer, the pluggable cross-attention modules can learn extensive interaction and semantic alignment.

### 3.3 Pre-training Objectives

We optimize M²-VLP by employing two types of pre-training objectives. On top of the uni-modal encoders, the model performs multi-grained contrastive learning. On top of the fusion encoder, the model is trained by masked language modeling and vision-language matching. Our goal is to achieve Vision-to-X languages alignment by learning multi-grained Vision-to-English alignments and English-to-X languages alignments. In this way, the model learns cascading alignments between Vision-English-X languages, with English serving as the aligning bridge.

*3.3.1 **Multi-grained contrastive learning**.* To cascadingly align unimodal representations before fusion, we propose a unified multi-grained contrastive learning to simultaneously align both of the English-to-Vision and English-to-X languages. For a batch of $N$ pairs $\{(A_i, B_i)\}_i^N$, $B_i$ is the positive sample for $A_i$, and the other $N - 1$ samples within the batch are negative samples. Specifically, $(A_i, B_i)$ refers to an image-text pair or a translation text pair. For each pair, we calculate the softmax-normalized $A$-to-$B$ similarity as:

$$\boldsymbol{p}_n^{A2B}(A) = \frac{exp(sim(A, B_n)/\tau)}{\sum_{n=1}^N exp(sim(A, B_n)/\tau)}. \tag{3}$$

Similarly, $B$-to-$A$ similarity is:

$$\boldsymbol{p}_n^{B2A}(B) = \frac{exp(sim(B, A_n)/\tau)}{\sum_{n=1}^N exp(sim(B, A_n)/\tau)}, \tag{4}$$

where $\tau$ is a learnable temperature parameter and the function $sim(\cdot)$ calculate the cosine similarity between $A$ and $B$. For the $i$-th pair, let $\boldsymbol{y}^{A2B}(A)$ and $\boldsymbol{y}^{B2A}(B)$ denote the ground-truth one-hot similarity, where negative pairs have a probability of 0 and the positive pair has a probability of 1. The contrastive loss of pair $(A, B)$ is defined as the cross-entropy $H$ between $\boldsymbol{p}$ and $\boldsymbol{y}$:

$$\mathcal{L}_{CL} = \frac{1}{2}\mathbb{E}_{(A,B)\sim D_{I,R,T}}[H(\boldsymbol{y}^{A2B}(A), \boldsymbol{p}^{A2B}(A)) + H(\boldsymbol{y}^{B2A}(B), \boldsymbol{p}^{B2A}(B))]. \tag{5}$$

In multi-grained contrastive, pair $(A_i, B_i)$ can be four types of data inputs. In the coarse-grained level, pair $(A_i, B_i)$ can be an embedded vector of an image-caption pair (from data stream $D_I$) or a textual translation pair (from data stream $D_T$). In fine-grained level, $(A_i, B_i)$ can be embedded vector of a region-token pair (from data stream $D_R$) or a token-token pair (from word align matrix $M$ of data stream $D_T$).

As shown on the right side of Figure 2, since no labeled word pairs are accessible, we utilize unsupervised word aligner (FastAlign[14]) to construct word align matrix $M$ for translation pairs. Specifically,

for every translation pair $(x, y)$ in data stream $D_T$, the element $\mathcal{M}_{i,j}$ of word align matrix $\mathcal{M}$ in $i$-th row and $j$-th column is defined as:

$$\mathcal{M}_{i,j} = \begin{cases} 1, & \text{if word } w_i^x \text{ is synonym of } w_j^y, \\ 0, & \text{otherwise,} \end{cases} \quad (6)$$

where $w_i^x$ is the $i$-th word of sentence $x$ and $w_j^y$ is the $j$-th word of sentence $y$. With the word align matrix $M$, the model is able to construct positive and negative samples in token-token contrastive learning.

Corresponding to the four types of multi-grained input data above, the contrastive loss functions can be denoted as $\mathcal{L}_{ICC}$ (image-caption contrastive), $\mathcal{L}_{RTC}$ (region-token contrastive), $\mathcal{L}_{SLC}$ (sentence-level contrastive), and $\mathcal{L}_{TLC}$ (token-level contrastive) respectively. Therefore, the overall objective function of multi-grained contrastive learning is:

$$\mathcal{L}_{MCL} = \lambda(\mathcal{L}_{ICC} + \mathcal{L}_{SLC}) + (1 - \lambda)(\mathcal{L}_{RTC} + \mathcal{L}_{TLC}), \quad (7)$$

where $\lambda$ is a scale hyper-parameter to balance coarse-grained contrastive and fine-grained contrastive.

*3.3.2 **Masked language modeling**.* Masked language modeling (MLM) is a simple but effective self-learning paradigm that has been proven in multiple domains [7, 13]. As shown in Figure 2, we design three variants of MLM: Vision MLM (VMLM) on image-caption pairs in $D_I$ and region-token pairs in $D_R$, cross-lingual MLM (xMLM) on multilingual texts in $D_M$, translation language modeling (TLM) on translation sentence pairs in $D_T$.

In VMLM, the model predicts the masked tokens based on both the image and unmasked tokens from texts. During the prediction process, the model needs to understand the masked word associated features of the image, which helps to learn cross-modal interaction. xMLM predicts the masked tokens in the multilingual texts. Previous studies [10, 11, 13] have indicated that xMLM training helps encode different languages into the shared embedding space, improving the model's ability in multilingual modeling and cross-language transfer. In TLM, the model can attend to both the English sentence and its translation to predict a masked token, thereby encouraging the alignment of representations across different languages.

In all three tasks, we randomly mask out the input tokens with a probability of 15%. Mathematically, we denote a masked text as $\hat{T}$, a masked parallel text pair as $(\hat{T}_X, \hat{T}_Y)$, and a masked image-text pair as $(\hat{T}_I, I)$. The complete MLM pre-training loss $\mathcal{L}_{MLM}$ can be obtained by summing up the $\mathcal{L}_{xMLM}$, $\mathcal{L}_{TLM}$, and $\mathcal{L}_{VMLM}$ together:

$$\begin{aligned} \mathcal{L}_{MLM} = & \, \mathbb{E}_{\hat{T} \sim D_M}[H(\mathbf{y}^{mlm}(\hat{T}), \mathbf{p}^{mlm}(\hat{T}))] \\ & + \mathbb{E}_{(\hat{T}_X, \hat{T}_Y) \sim D_T}[H(\mathbf{y}^{mlm}(\hat{T}_X, \hat{T}_Y), \mathbf{p}^{mlm}(\hat{T}_X, \hat{T}_Y))] \\ & + \mathbb{E}_{(\hat{T}_I, I) \sim D_{I,R}}[H(\mathbf{y}^{mlm}(\hat{T}_I, I), \mathbf{p}^{mlm}(\hat{T}_I, I))], \end{aligned} \quad (8)$$

where $\mathbf{p}^{mlm}$ is the predicted probability for a masked token, and $\mathbf{y}^{mlm}$ is a one-hot vocabulary distribution where the ground-truth token has a probability of 1.

*3.3.3 **Vision-language matching**.* In the vision-language matching (VLM) task, the model predicts whether a pair of an image and a text sequence is matched. In practice, the image and its corresponding caption are regarded as positive pairs, whereas the remaining examples in the batch are considered as negatives. To predict the matching probability $\mathbf{p}^{vlm}$, the model utilizes the output embedding of the [CLS] token from the fusion encoder as the joint representation of the image-text pair $(V, T)$. The VLM loss can be denoted as:

$$\mathcal{L}_{VLM} = \mathbb{E}_{(V,T) \sim D_{I,R}}[H(\mathbf{y}^{vlm}(V, T), \mathbf{p}^{vlm}(V, T))], \quad (9)$$

where the ground-truth label is represented as a one-hot vector $\mathbf{y}^{vlm}$.

Finally, the overall pre-training objective of M²-VLP is defined as:

$$\mathcal{L} = \mathcal{L}_{MCL} + \mathcal{L}_{MLM} + \mathcal{L}_{VLM} \quad (10)$$

## 4 Experiment

### 4.1 Pre-training Datasets

**Table 1: Statistics of annotations in image-text datasets.**

| Dataset | Images | Captions | Objects | Regions |
|---------|--------|----------|---------|---------|
| MSCOCO | 0.12M | 0.55M | 0.45M | — |
| Conceptual Captions | 3.33M | 3.33M | — | — |
| Visual Genome | 0.10M | — | 2.0M | 3.7M |

In pre-training stage, we consider 21 languages, including English, to cover all target languages in downstream datasets. To construct $D_M$, a subset of 400M multilingual sentences is sampled from the open-source dataset CC-100 [47], which is collected from the CommonCrawl[1] dump. As for $D_T$, we use 20M English-centric parallel sentences from WikiMatrix [37]. The specific language distribution in $D_M$ and $D_T$ is given in Appendix A.1. The data used to construct $D_I$ and $D_R$ is detailed in Table 1. From Conceptual Captions [39], MSCOCO [31], and Visual Genome [22], we use 3.9M image-caption pairs for $D_I$, along with 6.1M region-token pairs for $D_R$.

### 4.2 Pre-training Settings

For transformer modules, we adopt a base hidden size of 768 along with 12 heads for both self-attention and cross-attention. Following [26], we set $N_T = N_F = 6$ and $N_V = 12$. The vision encoder is initialized with [26]. The text encoder is initialized with the first six layers of XLMR [10], while the fusion encoder is initialized with the last six layers. Due to the absence of cross-attention modules in the XLMR model, we initialize the cross-attention layers with parameters of self-attention.

We optimize the model using the Adam optimizer for 240K steps. Each training batch consists of 512 image-text pairs, 512 region-token pairs, 2048 translation pairs, and 2048 multilingual sentences. The learning rate is scheduled with a linear decay with 24K warmup steps, where the peak learning rate is set as $1e-4$. The tokenizer of the XLMR is employed to tokenize the text. The maximum text sequence length of sentences in $D_M$, $D_T$, $D_I$, and $D_R$ are set to 64, 50, 35, and 35 respectively. The model is pre-trained at image resolution of $256 \times 256$ using $16 \times 16$ patch size. More details are given in Appendix A.2.

---

[1]https://commoncrawl.org/

**Table 2: Results of zero-shot cross-lingual transfer on IGLUE benchmark. The en indicates the results on the English test sets, while avg represents the average results for other target languages. IR and TR refer to image retrieval and text retrieval, respectively.**

| Model | VNLI | | VQA | | Reasoning | | Retrieval | | | | | | | |
| | XVNLI | | xGQA | | MaRVL | | xFlickr&CO | | | | WIT | | | |
| | | | | | | | IR | | TR | | IR | | TR | |
| | avg | en | avg | en | avg | en | avg | en | avg | en | avg | en | avg | en |
|---|---|---|---|---|---|---|---|---|---|---|---|---|---|---|
| mUNITER | 53.7 | 76.4 | 10.0 | 54.7 | 53.7 | 71.9 | 8.1 | 44.5 | 8.9 | 40.9 | 9.2 | 19.9 | 10.4 | 22.3 |
| xUNITER | 58.5 | 75.8 | 21.7 | 54.8 | 54.6 | 71.6 | 14.0 | 38.5 | 13.5 | 32.1 | 8.7 | 16.7 | 9.8 | 18.5 |
| UC$^2$ | 62.1 | 76.4 | 29.4 | 55.2 | 57.3 | 70.6 | 20.3 | 37.4 | 17.9 | 34.6 | 7.8 | 17.9 | 9.1 | 19.7 |
| M$^3$P | 58.3 | 76.9 | 28.2 | 53.8 | 56.0 | 68.2 | 12.9 | 31.4 | 11.9 | 24.6 | 8.1 | 15.5 | 10.0 | 15.3 |
| WS-mVLP | 69.5 | 79.7 | 42.1 | 57.4 | 62.1 | 75.3 | 59.8 | 86.6 | 58.7 | **91.7** | 36.3 | 56.0 | 36.6 | 56.2 |
| Ours | **71.1** | **80.3** | **46.6** | **58.0** | **64.6** | **80.3** | **71.4** | **91.9** | **70.7** | 84.5 | **46.6** | **71.2** | **45.1** | **71.0** |

## 4.3 Downstream Tasks

We conduct comprehensive experiments to evaluate the proposed model across two settings:

**Zero-shot cross-lingual transfer in vision-language tasks:** To assess the cross-lingual transferability of the proposed model in vision-language tasks, we evaluate M$^2$-VLP on a recently released IGLUE [5] benchmark. For all tasks in this scenario, we follow a zero-shot cross-lingual transfer setting in which the model is trained only in English and evaluated directly in other languages. The five tasks of the IGLUE benchmark include:

- **XVNLI** Cross-lingual Visual Natural Language Inference (XVNLI) [5] requires the model to predict if a text-hypothesis 'entails', 'contradicts', or is 'neutral' to an image-premise. The dataset is collected by combining SNLI [4] with its multi-modal [27] and cross-lingual [1] counterparts.
- **xGQA** The Cross-lingual Grounded Question Answering (xGQA) task [35] is collected by manually translating the GQA validation set into 7 languages, while training data are sourced from the English training set of GQA. It requires a model to answer several types of structured questions about an image.
- **MaRVL** The Multicultural Reasoning over Vision and Language (MaRVL) dataset [32] requires the model to determine whether a textual description is true or false about a pair of images. The task involves comparing two visual representations and reasoning about the information in the textual description.
- **xFlickr&CO** The dataset [5] is collected by combining 1000 images from Flickr30K [49] and MSCOCO [31] respectively. It is a retrieval task with two subtasks: image-to-text retrieval (TR) and text-to-image retrieval (IR). The image captions encompass a total of eight languages, including English.
- **WIT** Wikipedia-based Image Text dataset (WIT) [40] is collected from Wikipedia in 108 languages. Similar to xFlickr&CO, the tasks consist of retrieving the correct image given a text (IR) and vice versa (TR). The English training set consists of 500K captions, and the evaluation set comprises 10 languages, each containing at least 500 image-text pairs.

**Multilingual fine-tuning in vision-language tasks:** To evaluate the multilingual ability of the proposed model, we conduct experiments on image-text retrieval tasks. We consider two settings, one is to train and test each language independently, and the other is to combine all languages into a unified training set and test on different languages. Following [30], We use multilingual extensions of MSCOCO [31] and Multi30K [15], both of which offer multilingual annotations for training.

- **Multi30K** This dataset extends Flickr30K [49] from English to German, French and Czech. It consists of 31,783 images, each paired with five captions in English and German, and one caption in French and Czech. We follow the data split in [30].
- **MSCOCO** This dataset contains 123,287 images and provides five captions per image in English. STAIR [48] extends the original MSCOCO with 820K Japanese captions. Moreover, COCO-CN [28] extends MSCOCO with Chinese captions for nearly 20K images. For English and Japanese, we follow the data split in [21]. As for Chinese, we use the original COCO-CN split.

In fine-tuning stage, we follow the task-specific hyper-parameters in the baseline model [30]. We report accuracy for XVNLI, xGQA, and MaRVL. For retrieval tasks, we use mean Recall (mR) as our evaluation metric, which is an averaged score of R@1, R@5, and R@10 on image-to-text retrieval and text-to-image retrieval tasks.

## 4.4 Baseline Models

We compare our model with recent competitive multilingual VLP models: mUNITER [32], xUNITER [32], UC$^2$ [52], M$^3$P [34], and WS-mVLP [30]. Among the baselines, mUNITER and xUNITER, which are initialized from mBERT and XLM-R respectively, employ the UNITER architecture and are pre-trained with MLM on both cross-lingual texts and English image-text pairs. M$^3$P implements explicit alignment by employing a code-switching strategy, replacing English words in image-text pairs with their respective synonyms in the target languages. UC$^2$ utilizes translation engines to translate existing English-only image-text pairs into other five

**Table 3: Results of multilingual image-text retrieval on Flickr30K and MSCOCO.**

| Model | Flickr30K | | | | MSCOCO | | |
|---|---|---|---|---|---|---|---|
| | en | de | fr | cs | en | zh | ja |
| *English only Finetune* | | | | | | | |
| UC² | 87.2 | 74.9 | 74.0 | 67.9 | 88.1 | 82.0 | 71.7 |
| M³P | 87.4 | 58.5 | 46.0 | 36.8 | 88.6 | 53.8 | 56.0 |
| WS-mVLP | 94.9 | 84.4 | 86.1 | 77.2 | 89.6 | 83.3 | 73.1 |
| Ours | **95.3** | **88.6** | **90.3** | **85.9** | **90.7** | **88.9** | **83.3** |
| *Single Language Finetune* | | | | | | | |
| UC² | 87.2 | 83.8 | 77.6 | 74.2 | 88.1 | 84.9 | 87.3 |
| M³P | 87.4 | 82.1 | 67.3 | 65.0 | 88.6 | 75.8 | 80.1 |
| WS-mVLP | 94.9 | 92.5 | 92.4 | 91.0 | 89.6 | **92.5** | 90.4 |
| Ours | **95.3** | **93.2** | **92.6** | **91.6** | **90.7** | 91.9 | **91.2** |
| *All-Language Finetune* | | | | | | | |
| UC² | 88.2 | 84.5 | 83.9 | 81.2 | 88.1 | 89.8 | 87.5 |
| M³P | 87.7 | 82.7 | 73.9 | 72.2 | 88.7 | 86.2 | 87.9 |
| WS-mVLP | **95.3** | 93.6 | **93.8** | 92.4 | 90.4 | 92.6 | 90.0 |
| Ours | 95.2 | **94.0** | **93.8** | **92.8** | **90.8** | **92.7** | **91.6** |

languages and a visual-conditioned translation language modeling objective is introduced. WS-mVLP unifies cross-lingual and cross-modal pre-training within a weakly supervised framework, achieving remarkable results on the IGLUE benchmark. The experimental results for the baseline models are sourced from [30].

## 4.5 Main Results

In order to comprehensively evaluate the multilingual performance of the model, we conduct experiments under the settings of zero-shot cross-lingual transfer and multilingual fine-tuning:

**Cross-lingual Transfer**    As shown in Table 2, M²-VLP achieves superior zero-shot cross-lingual performance across various vision-language tasks. Specifically, compared to the best baseline WS-mVLP, our model achieves an average accuracy improvement of 6.9% on zero-shot cross-lingual vison-language understanding tasks including XVNLI, xGQA, and MaRVL. For retrieval tasks, M²-VLP outperforms all compared models by a substantial margin in cross-lingual settings. On the English test sets, our method exhibits the best performance on all tasks except for TR in xFlickr&CO. These results suggest that our multi-grained vision-language pre-training can learn better alignment between image and multilingual texts.

**Multilingual Fine-tune**    As shown in Table 3, We conduct experiments on the multilingual extensions of Flickr30K and MSCOCO under three settings, which are zero-shot English-only fine-tuning, target language fine-tuning, and all-language fine-tuning. Compared to the baseline models, M²-VLP shows improvement across different languages. Under the setting of all-language fine-tuning, the improvement is not prominent. In contrast, our model exhibits more improvements under the cross-lingual transfer setting, possibly attributed to the additional fine-grained alignment between different languages.

**Table 4: Results of ablation studies. TLC is the abbreviation for token-level contrastive, RTC for region-token contrastive, and CS for code-switching.**

| Model | XVNLI | xGQA | MaRVL | xFlickr&CO | | WIT | |
|---|---|---|---|---|---|---|---|
| | | | | IR | TR | IR | TR |
| Ours | **71.1** | **46.6** | **64.6** | **71.4** | **70.7** | **46.6** | **45.1** |
| *w/o TLC* | 70.7 | 45.8 | 64.4 | 71.3 | 69.5 | 45.4 | 43.3 |
| *w/o RTC* | 70.1 | 46.4 | 64.1 | 70.3 | 68.1 | 45.6 | 43.7 |
| *w/o CS* | 69.7 | 42.0 | 62.8 | 69.5 | 69.1 | 46.2 | 43.7 |

## 4.6 Ablation Study

We conduct ablation studies to investigate the effect of different components in the proposed model. We discuss the following three model variants:

- *w/o TLC*: This variant removes the token-level contrastive objective $\mathcal{L}_{TLC}$, which means that the model aligns cross-lingual texts with only sentence-level contrastive.
- *w/o RTC*: In this setting, we remove region-token contrastive objective $\mathcal{L}_{RTC}$, which means the model only performs coarse-grained cross-modal aligning with image-caption contrastive.
- *w/o CS*: This variant removes code-switching augmentation module (in section 3.1) for $D_I$ and $D_R$.

The results are shown in table 4. In general, all three modules have contributed to the model performance. Firstly, TLC showed significant improvement in xGQA and WIT tasks, indicating their sensitivity to cross-lingual fine-grained alignment. Secondly, RTC demonstrates more noticeable improvement on the XVNLI and xFlickr&CO datasets, indicating that token-region alignment may be more important for these two tasks. Surprisingly, the CS module has contributed the most to performance improvement. Previous studies have demonstrated that randomly replacing English words with their synonyms is a simple yet effective method that can promote word-level cross-lingual alignment. Compared to retrieval tasks, CS has shown greater improvement in XVNLI, xGQA, and MaRVL tasks, achieving an average improvement of 2.6%. These results demonstrate the importance of the proposed fine-grained alignment training for cross-lingual vision-language tasks.

## 4.7 Visualization

**Cross-attention Visualization**    To obtain an intuitive comprehension of the proposed model, we use Grad-CAM [38], a commonly used "visual explanation" toolkit, to generate cross-attention location maps for the last layer of the fusion encoder. As shown in Figure 3, words from different languages that convey the same meaning can activate to corresponding regions in the image. It indicates that our model can effectively transfer the cross-modal alignment knowledge learned from English to other languages. More examples of cross-attention visualization are shown in Appendix B.2.

**Cross-lingual Aligning**    To explore whether our model tends to learn fine-grained cross-lingual alignment, we employ t-SNE [43] to visualize the distances between representations of translation word pairs. As Figure 4 shows, we sample 15 English-X translation word

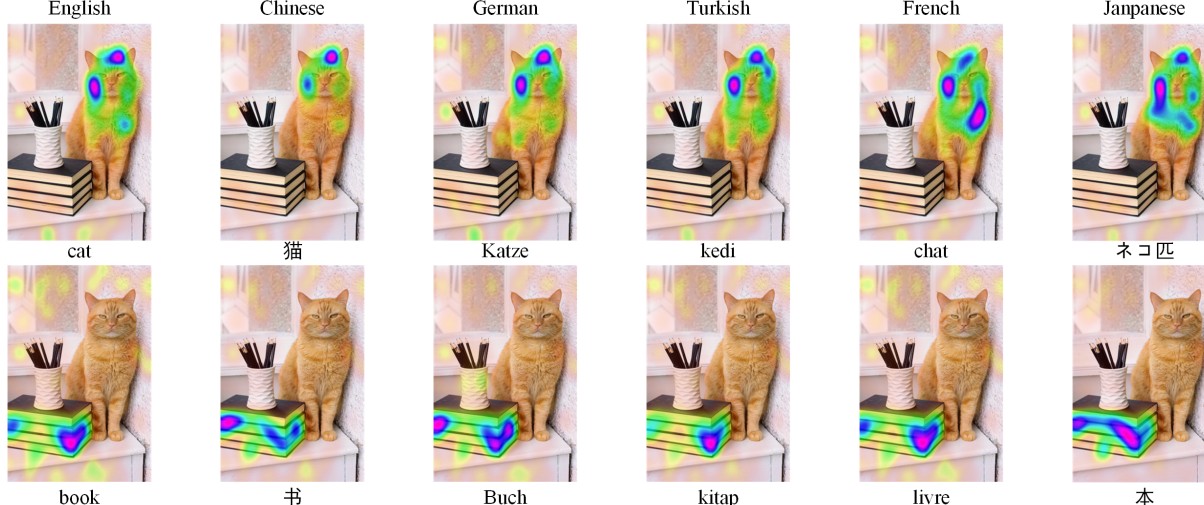

**Figure 3: Grad-CAM visualization of the cross-attention between regions and corresponding words across different languages.**

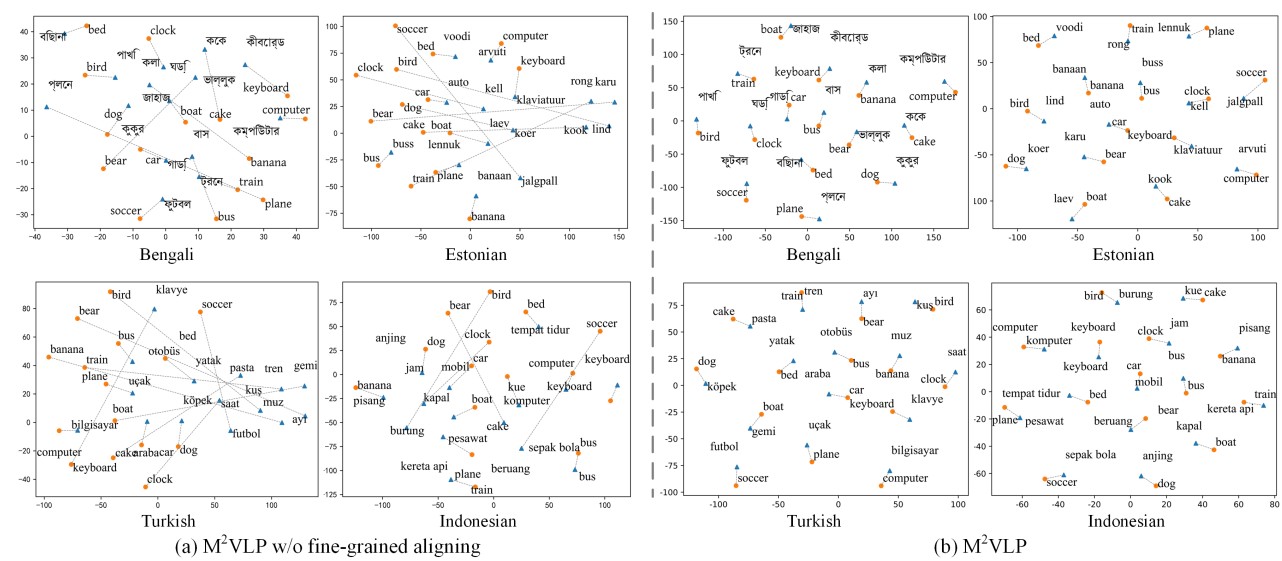

(a) M²VLP w/o fine-grained aligning                                                        (b) M²VLP

**Figure 4: The tSNE visualization of the word representations. Each word pair is connected by a grey dotted line.**

pairs (excluding stop words) from $D_I$ for four low-resource target languages. The hidden states of the last transformer layer in the fusion encoder serve as the word representations. Figure 4(a) shows the t-SNE visualization of our model without fine-grained aligning (without $\mathcal{L}_{RTC}$ and $\mathcal{L}_{TLC}$)). We observe that only a few word pairs are properly aligned and words from the same language are more likely to gather in the hidden space. As for fully trained M²VLP in Figure 4(b), even in the case of low-resource languages, the results show that they achieve significant token-level semantic alignment with English, which leads to better performance in cross-lingual transfer. The cross-lingual aligning visualization of other languages is presented in Appendix B.3.

## 5 Conclusion

In this paper, we propose M²-VLP to perform multi-grained multilingual vision-language pre-training. Our approach focuses on enhancing fine-grained alignment in two dimensions: cross-lingual and cross-modal. Specifically, we introduce two novel training objectives, RTC and TLC, to enhance semantic alignment between image regions and corresponding phrases, as well as words in different languages. Then we perform multi-grained multilingual vision-language pre-training by unifying training paradigms from different granular levels. Experimental results demonstrate that M²-VLP outperforms previous state-of-the-art models on various cross-lingual vision-language benchmarks.

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

# A Additional Implementation Details

## A.1 Pre-training data

For parallel text data stream $D_T$, we use all data of target languages in WikiMatrix, totaling 20M English-centric parallel sentence pairs. The data size distribution of training samples in different languages is shown in Figure 5. As for multilingual data stream $D_M$, due to the significant differences in the number of training samples for different languages, we follow [10] to sample 400M sentences according to a multinomial distribution. We denote $n_i$ is the number of sentences in $L_i$ in the full dataset, and the sampling probabilities $\{q_i\}_{i=1,2,\dots,N}$ of language $L_i$ can be obtained by:

$$ q_i = \frac{p_i^\alpha}{\sum_{j=1}^N p_j^\alpha} \quad \text{with} \quad p_i = \frac{n_i}{\sum_{k=1}^N n_k} \tag{11} $$

We set $\alpha = 0.3$ following [30]. It increases the sampling probability of low-resource languages and alleviates the bias towards high-resource languages. The language distribution in $D_M$ is illustrated in Figure 6. We use ISO-639 language codes to abbreviate all languages.

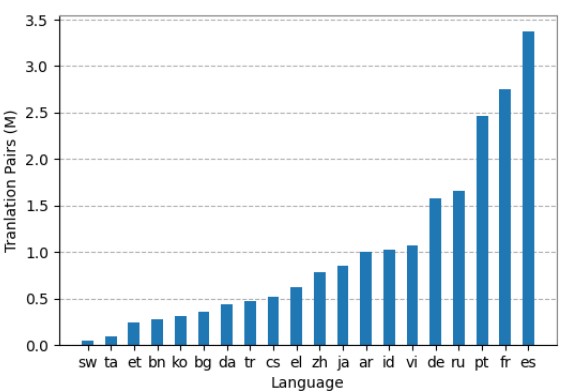

Figure 5: Data size of different languages in $D_T$.

## A.2 Hyper-parameters

Our model consists of approximately 377M parameters, in which the word embeddings for the large vocabulary account for 200M. We apply RandAugment [12] to the input images. To prevent overfitting, a dropout rate of 0.1 is utilized, along with a weight decay of 0.2 in the optimizer. $\lambda$ is set to 0.5 in Equation 7. In VMLM, xMLM, and TLM, following [10], we keep 15% masking ratio, the replacements are 10% random tokens, 10% unchanged, and 80% [MASK] tokens. Following [26], we increase the image resolution to 384 × 384 and interpolate the positional embeddings of image patches during the fine-tuning stage. All experiments are conducted on 8 NVIDIA RTX 4090 GPUs.

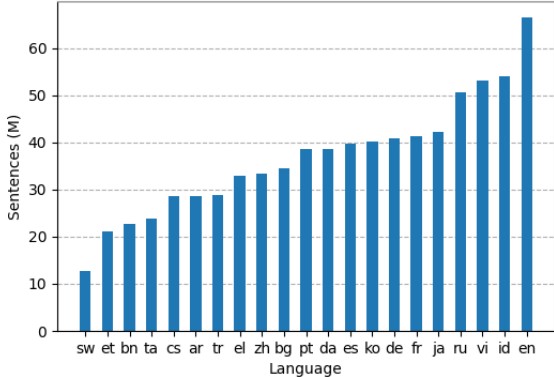

Figure 6: Data size of different languages in $D_M$ after sampling.

Table 5: Language-specific results of zero-shot cross-lingual transfer on XVNLI.

| Model | ar | es | fr | ru | avg |
|---|---|---|---|---|---|
| UC$^2$ | 56.2 | 57.5 | 69.7 | 64.9 | 62.1 |
| M$^3$P | 55.3 | 58.9 | 56.4 | 62.5 | 58.3 |
| WS-mVLP | 66.3 | 69.5 | 71.7 | **70.4** | 69.5 |
| Ours | **69.0** | **72.6** | **73.2** | 69.7 | **71.1** |

Table 6: Language-specific results of zero-shot cross-lingual transfer on xGQA.

| Model | bn | de | id | ko | pt | ru | zh | avg |
|---|---|---|---|---|---|---|---|---|
| UC$^2$ | 20.0 | 42.9 | 28.7 | 21.4 | 30.4 | 31.0 | 31.2 | 29.4 |
| M$^3$P | 18.6 | 33.4 | 32.5 | 25.1 | 31.4 | 27.5 | 28.7 | 28.2 |
| WS-mVLP | 31.9 | 48.7 | 45.3 | 39.1 | 47.0 | 39.0 | 43.4 | 42.1 |
| Ours | **41.2** | **50.3** | **49.3** | **44.1** | **49.4** | **45.3** | **46.4** | **46.6** |

# B Additional Results and Analysis

## B.1 Language-specific results

Language-specific results in all tasks are illustrated in Table 5,6,8,7, and 9. Overall, we observe that our model exhibits more improvement in low-resource languages compared to high-resource languages. This indicates that fine-grained semantic alignment can partially compensate for performance bottlenecks caused by limited training resources. In the XVNLI task, our model achieves the best results in all languages except Russian. In the xGQA task, M$^2$-VLP outperforms the baseline models in all languages and achieves a 4.5% improvement on average. In the MaRVL task, our model achieves the best performance in all five target languages, with an average improvement of 2.5%. In the retrieval tasks of xFlickr&CO and WIT, our model performs excellently in both text retrieval and image retrieval. The above results demonstrate that our model has good generalization capabilities across different languages.

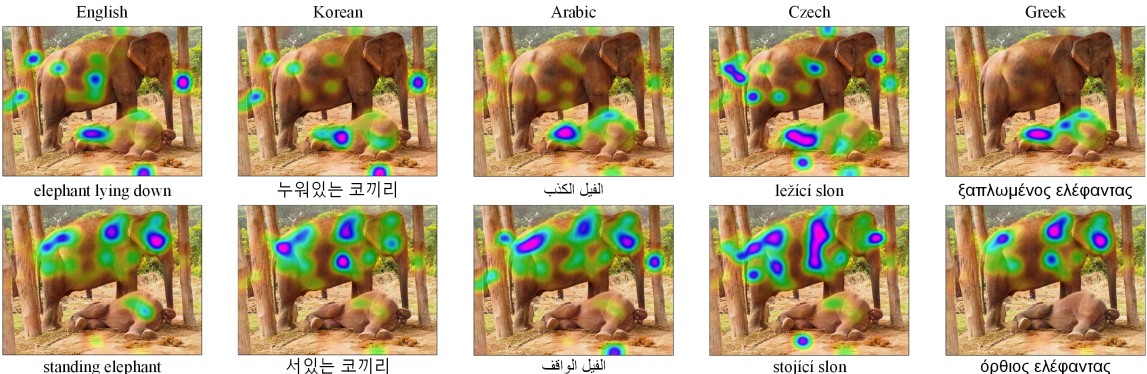

**Figure 7: Grad-CAM visualization of the cross-attention. We evaluate the model's ability to distinguish colors in different languages.**

**Figure 8: Grad-CAM visualization of the cross-attention. We evaluate the model's ability to distinguish animal postures in different languages.**

**Table 7: Language-specific results of zero-shot cross-lingual transfer on xFlickr&CO.**

| Model | de | | es | | id | | ja | | ru | | tr | | zh | | avg | |
|---|---|---|---|---|---|---|---|---|---|---|---|---|---|---|---|---|
| | IR | TR | IR | TR | IR | TR | IR | TR | IR | TR | IR | TR | IR | TR | IR | TR |
| UC² | 28.6 | 23.9 | 16.0 | 15.3 | 14.6 | 13.6 | 24.3 | 22.4 | 20.0 | 16.8 | 7.2 | 7.0 | 31.6 | 26.3 | 20.3 | 17.9 |
| M³P | 13.4 | 11.9 | 13.4 | 12.2 | 13.2 | 12.1 | 10.3 | 9.7 | 16.0 | 14.5 | 7.8 | 8.4 | 16.5 | 14.8 | 12.9 | 11.9 |
| WS-mVLP | 58.2 | 57.2 | 69.6 | 68.7 | 62.7 | 60.6 | 49.8 | 48.2 | 63.2 | 62.6 | 50.8 | 50.8 | 64.2 | 63.2 | 59.8 | 58.7 |
| Ours | **67.4** | **67.2** | **77.5** | **77.6** | **71.1** | **70.1** | **65.9** | **64.8** | **75.4** | **75.3** | **69.7** | **68.2** | **73.1** | **72.0** | **71.4** | **70.7** |

**Table 8: Language-specific results of zero-shot cross-lingual transfer on MaRVL.**

| Model | id | sw | ta | tr | zh | avg |
|---|---|---|---|---|---|---|
| UC² | 56.7 | 52.6 | 60.5 | 56.7 | 59.9 | 57.3 |
| M³P | 56.5 | 55.7 | 56.0 | 56.8 | 55.0 | 56.0 |
| WS-mVLP | 65.3 | 58.7 | 60.3 | 65.3 | 60.6 | 62.1 |
| Ours | **65.7** | **60.2** | **61.6** | **71.2** | **64.1** | **64.6** |

## B.2 Additional cross-attention visualization

To further validate whether the model can capture fine-grained image-text alignment information, we conduct additional visualization analysis of the cross-attention layers for two images. In Figure 7, we input "brown horse" and "black horse" in different languages to our model. The heatmap regions indicate model's ability to distinguish color differences well, even in low-resource languages. In Figure 8, we analyze the model's ability to discern the postures of elephants. Surprisingly, the cross-attention layers of our model can effectively model the differences. The above visualization analysis

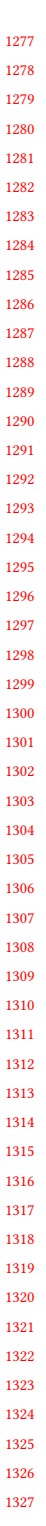

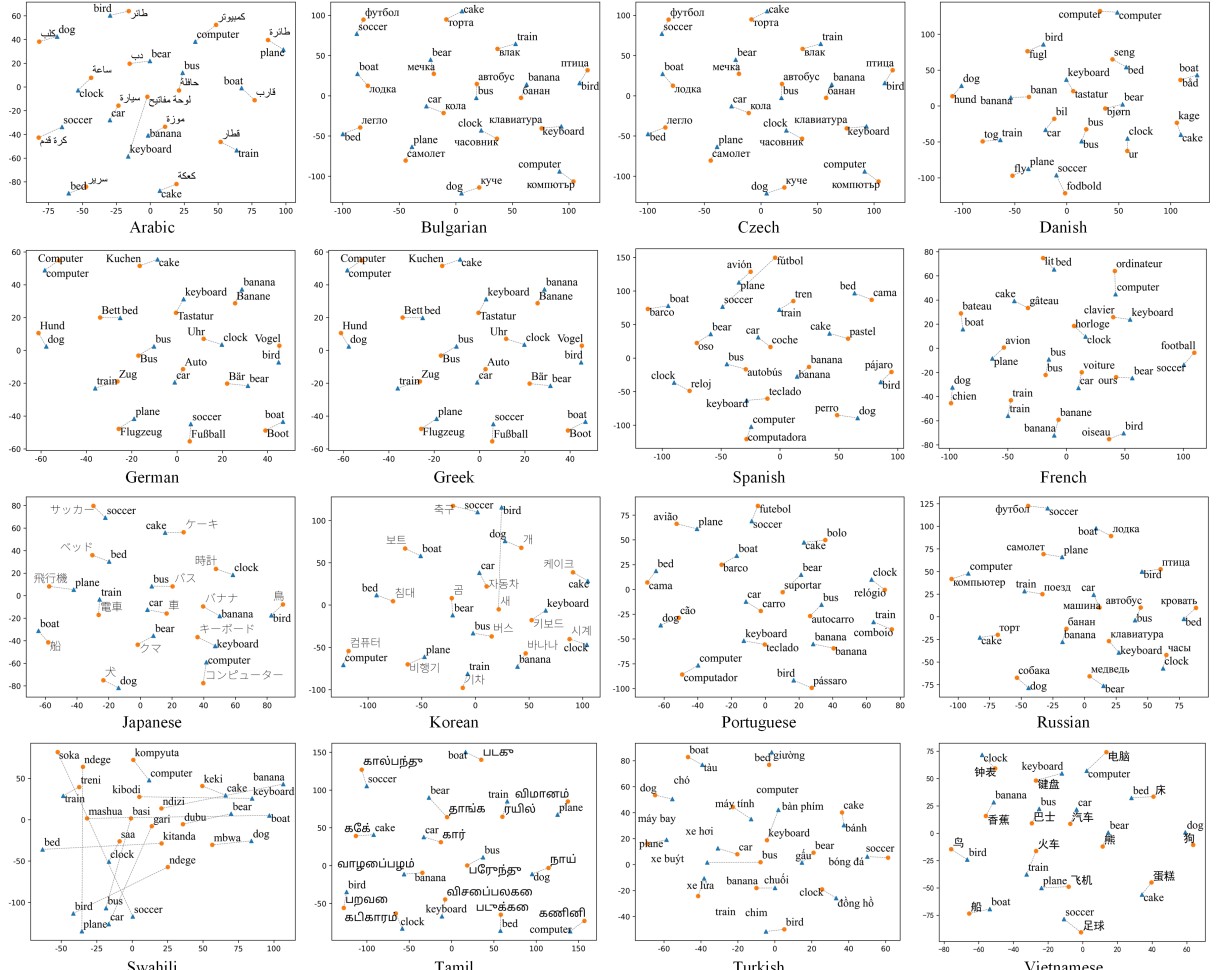

**Figure 9: Additional tSNE visualization of the word representations.**

**Table 9: Language-specific results of zero-shot cross-lingual transfer on WIT.**

| Model | ar IR | ar TR | bg IR | bg TR | da IR | da TR | el IR | el TR | et IR | et TR | id IR | id TR | ja IR | ja TR | ko IR | ko TR | tr IR | tr TR | vi IR | vi TR | mean IR | mean TR |
|-------|------|------|------|------|------|------|------|------|------|------|------|------|------|------|------|------|------|------|------|------|------|------|
| UC2 | 6.6 | 8.3 | 8.8 | 7.7 | 9.4 | 10.4 | 8.8 | 11.6 | 4.7 | 6.0 | 9.9 | 11.5 | 9.8 | 10.8 | 4.3 | 5.7 | 7.5 | 8.8 | 8.5 | 9.9 | 7.8 | 9.1 |
| M3P | 8.9 | 8.3 | 8.8 | 9.8 | 9.4 | 11.8 | 9.7 | 12.0 | 5.4 | 8.2 | 8.7 | 10.9 | 7.0 | 8.4 | 6.1 | 7.1 | 6.5 | 10.6 | 10.8 | 12.7 | 8.1 | 10.0 |
| WS-mVLP | 37.3 | 37.8 | 30.8 | 31.4 | 41.8 | 40.7 | 37.7 | 37.3 | 26.5 | 26.9 | 47.1 | 44.0 | 31.9 | 33.3 | 25.6 | 26.0 | 36.1 | 40.8 | 48.1 | 47.3 | 36.3 | 36.6 |
| Ours | **48.2** | **46.9** | **42.6** | **41.1** | **51.7** | **49.7** | **48.6** | **46.3** | **37.4** | **35.0** | **55.5** | **53.6** | **39.5** | **38.2** | **33.9** | **35.1** | **49.7** | **46.7** | **59.1** | **58.2** | **46.6** | **45.1** |

demonstrates the model's ability to model fine-grained information in cross-lingual cross-modal scenarios.

## B.3 Additional tSNE visualization

As shown in Figure 9, we also conduct tSNE visualization analysis of word alignment for other languages. The translations of 15 English words are obtained from Google Translate[2]. We observe that most languages achieve good word alignment, with relatively small

distances between synonyms. From the perspective of language families, languages (e.g. German, French, etc.) that belong to the same Indo-European language family as English can align better. However, only a few word pairs are properly aligned in Swahili. From Figure 5 and Figure 6, we can see that the training data for the Swahili language is the least in both $D_T$ and $D_M$. We infer that the insufficient amount of training data has prevented the model from learning fine-grained word alignment between Swahili and English.

[2]https://translate.google.com/

