# OpenReview forum: "M$^2$-VLP: Enhancing Multilingual Vision-Language Pre-Training via Multi-Grained Alignment"
_ACM.org/TheWebConf/2025/Conference — WWW 2025 Poster_

### Official Review · Reviewer_h18i · 2024-10-31

**Novelty:** 6
**Technical Quality:** 7

**Review:**

Evaluation of "M2-VLP: Enhancing Multilingual Vision-Language Pre-Training via Multi-Grained Alignment"

The paper presents a novel approach to multilingual vision-language pre-training (mVLP) by introducing the M2-VLP model, which focuses on multi-grained alignment across languages and modalities. The quality of the work is high.

The paper is well-structured and clear in its presentation. The introduction effectively sets the stage by outlining the limitations of current mVLP methods and the motivation behind the proposed M2-VLP model. The methodology is explained in detail, making it understandable for readers. The results are presented clearly, with tables and visualizations that effectively communicate the performance of M2-VLP compared to other models.

The originality of this work lies in its multi-grained approach to alignment, which includes both cross-lingual and cross-modal interactions at different semantic granular levels. The introduction of tasks like region-token contrastive learning and token-level contrastive learning is innovative and addresses a gap in the field of mVLP.

The significance of this work is substantial. It pushes the boundaries of mVLP by enhancing the learning of joint representations across different languages and modalities.

A suggestion: More information can be provided on specific experimental details, such as hyperparameter selection and data preprocessing steps, to enhance the reproducibility of the experiments.

**Questions:**

1.Does the M2-VLP model have potential commercial applications? For example, in areas such as advertising and recommendation systems.
2.Does the commercialization of the model face specific legal or ethical challenges, such as copyright issues and user privacy protection?

**Reviewer Confidence:**

4: The reviewer is certain that the evaluation is correct and very familiar with the relevant literature

**Scope:**

4: The work is relevant to the Web and to the track, and is of broad interest to the community

---

### Official Review · Reviewer_Mn88 · 2024-11-18

**Novelty:** 4
**Technical Quality:** 6

**Review:**

# Summary
- The M2-VLP model was developed to improve multilingual vision-language pre-training.

- Unlike existing models that perform coarse-grained semantic alignment, M2-VLP learns fine-grained correlations through multi-grained alignment.

- It utilizes translation pairs to learn both sentence- and word-level alignment, capturing fine-grained semantic associations between images and text.

- Demonstrates outstanding performance in various vision-language tasks, outperforming competitive models in 20 languages in zero-shot and multilingual settings.

# Strengths

- Captures deeper semantic associations through fine-grained alignment performance.

- Achieves high performance across diverse languages and tasks.

- Provides consistent alignment performance across multiple modalities and languages through an integrated contrastive learning framework.

# Weaknesses
- High computational cost due to the complex training structure

- There is insufficient detail regarding the specific pretrained datasets. A thorough description is required of how each dataset is structured and how it is employed in the computation of the respective loss functions.

- The individual loss functions (sentence-word level, sentence-another language sentence, image-sentence) are already publicly available. It seems that this work merely combines these losses without offering any novel contributions.

- A detailed explanation of prior studies, along with a contrasting description of the proposed contributions of this paper, is required. As it stands, it is challenging to clearly identify the novelty of this research.

**Questions:**

- Could you provide a more detailed explanation of how the Unsupervised Word Aligner is trained as presented in Figure 2?

- It would be beneficial to examine the weaknesses of the study and provide additional feedback. While the paper demonstrates impressive engineering capabilities, it is unclear whether it presents any academically novel findings.

- It is unclear whether this research falls within the scope of The Web Conference. Could you provide an explanation of how this study is relevant to the web, supporting its submission to this conference?

**Reviewer Confidence:**

3: The reviewer is confident but not certain that the evaluation is correct

**Scope:**

2: The connection to the Web is incidental, e.g., use of Web data or API

---

### Official Review · Reviewer_EBrg · 2024-11-19

**Novelty:** 3
**Technical Quality:** 4

**Review:**

The paper "Multi-grained Multilingual Vision-Language Pre-training (M²-VLP)" introduces an innovative approach to multilingual Vision-Language Pre-training (mVLP). While existing models focus on coarse-grained semantic alignment, the M²-VLP model distinguishes itself by capturing fine-grained correlations between languages and modalities. While M²-VLP offers innovative contributions in fine-grained semantic alignment and demonstrates effectiveness across several vision-language tasks, its comparative evaluations with state-of-the-art visual grounding works are lacking. Addressing these comparisons could further bolster the paper's claims and provide a clearer picture of its advancements relative to the current leading methods in visual grounding.

**Questions:**

The paper claims to be "the first known effort to explicitly learn multi-grained cross-lingual and cross-modal alignment in multilingual vision-language pre-training." However, there are several visual grounding works in 2024, such as Qwen2 VL, which use bounding boxes to provide multi-grained information for multilingual pre-training. Can you provide more comparisons with these contemporary visual grounding methods, as this would offer a clearer picture of your model's relative performance and innovations?

Some paper about visual grounding for pre-training can be found as follows:
https://arxiv.org/pdf/2409.12191
https://pubmed.ncbi.nlm.nih.gov/38109234/
https://arxiv.org/pdf/2212.09737

**Reviewer Confidence:**

4: The reviewer is certain that the evaluation is correct and very familiar with the relevant literature

**Scope:**

2: The connection to the Web is incidental, e.g., use of Web data or API

---

### Official Review · Reviewer_7mCJ · 2024-11-22

**Novelty:** 3
**Technical Quality:** 4

**Review:**

The motivation of this paper is clear, aiming to address cross linguistic alignment and cross modal alignment. It employs contrastive learning in conjunction with four tasks and utilizes a diverse range of datasets
advantage:
The importance of the research question: The paper focuses on the fine-grained alignment problem of multilingual visual language pre training, which has a certain degree of cutting-edge significance.
The experiment covers a wide range of cross linguistic and cross modal tasks, with detailed data analysis.
Method innovation: Introducing multi granularity contrastive learning and multiple pre training objectives, attempting to compensate for the shortcomings of existing methods.
Disadvantages and issues:
Lack of innovation:
The method of the paper has undergone minor modifications based on existing models such as WS mVLP, M3P, etc., mainly as a simple extension of the contrastive learning objective, and has not demonstrated breakthrough theoretical or technical contributions.
Although the concept of "multi granularity alignment" is novel, its practical implementation (such as word level comparison, region alignment, etc.) highly overlaps with existing research and fails to demonstrate uniqueness.
Unclear method description:
The implementation details of multi granularity contrastive learning, such as Token Level Contrastive and Region Token Contrastive, are described vaguely and lack clear mathematical derivation and intuitive explanation.
The pre training objectives (such as VMLM and TLM) in the paper did not show significant differences compared to related works, and the logic of the discussion is not clear enough.

**Questions:**

Is there any separate tuning of hyperparameters for different tasks in the experiment? If not, is a unified hyperparameter selection not friendly enough for certain tasks?
Does data distribution affect the performance of the model? Does the weight of low resource languages in training data affect cross language transfer ability?
Is the proposed method still applicable on large-scale data? Has the demand for training time and computing resources significantly increased? How to balance performance improvement and resource consumption?
The article did not mention a large language model. Can we consider using a large model to achieve better results?

**Reviewer Confidence:**

3: The reviewer is confident but not certain that the evaluation is correct

**Scope:**

3: The work is somewhat relevant to the Web and to the track, and is of narrow interest to a sub-community

---

### Official Review · Reviewer_QdAx · 2024-12-03

**Novelty:** 5
**Technical Quality:** 5

**Review:**

The paper addresses the area of multilingual vision-language pre-training (mVLP), the proposed M2-VLP model introduces multi-grained alignment, a novel approach that integrates fine-grained token-token, region-token, and sentence-level alignments, which is a notable advancement over existing coarse-grained methods.
Specifically, the unified multi-grained contrastive learning framework is well-motivated and effectively integrates cross-lingual and cross-modal alignments without requiring multilingual image-text datasets.
The introduction of token-level and region-level contrastive losses enhances the model’s ability to capture detailed semantic correlations.

**Questions:**

1. While the ablation studies highlight the contributions of key components (e.g., token-level contrastive loss), further explanation of the interaction between these components could enhance clarity.
For example, how do the token-level and region-level contrastive objectives jointly contribute to tasks involving complex reasoning?
2. Although the paper compares with some baselines such as WS-mVLP and UC2, additional analyses on the trade-offs (e.g., computational costs versus performance) would provide a more balanced perspective.
3. In constructing region-token pairs, how are ambiguous or overlapping annotations handled? Could this lead to inconsistencies in the fine-grained alignment?
4. I am wondering how the performance varies when one of the data streams (multilingual text, parallel text, and monolingual image-text streams) is excluded. Are all streams equally important for different downstream tasks?
5. Could the authors clarify whether the token-level and region-token alignment is equally effective across low-resource and high-resource languages?
6. In xGQA, the model performs well, but visual reasoning is typically challenging for fine-grained approaches. Could the authors provide examples of where the model succeeds or fails in reasoning-based tasks?

**Reviewer Confidence:**

3: The reviewer is confident but not certain that the evaluation is correct

**Scope:**

3: The work is somewhat relevant to the Web and to the track, and is of narrow interest to a sub-community